# Current Molecular Profile of Gastrointestinal Stromal Tumors and Systemic Therapeutic Implications

**DOI:** 10.3390/cancers14215330

**Published:** 2022-10-29

**Authors:** Maria Cecilia Mathias-Machado, Victor Hugo Fonseca de Jesus, Leandro Jonata de Carvalho Oliveira, Marina Neumann, Renata D’Alpino Peixoto

**Affiliations:** 1Centro Paulista de Oncologia (Oncoclínicas), São Paulo 04538-132, Brazil; 2OC Oncoclínicas—Unimed Grande Florianópolis, Florianópolis 88015-020, Brazil

**Keywords:** gastrointestinal stromal tumor, *KIT*, *PDGFRA*, wild-type GIST, SDH-deficient

## Abstract

**Simple Summary:**

Gastrointestinal stromal tumors (GISTs) are malignant mesenchymal tumors classified primarily as soft tissue sarcomas and characteristically arise from the intestinal pacemaker cells of Cajal responsible for the gastrointestinal tract motility. They comprise a heterogenous group of tumors due to a variety of molecular alterations, mostly *KIT* (60–70%) or *PDGFRA* (10–15%) mutations, but also more rare alterations, including inactivation of the *NF1* gene, mutations in the succinate dehydrogenase (*SDH*), *BRAF*, and *RAS* genes, and also gene fusions. Here, we review the most recent data on the molecular profile of GISTs and highlight systemic therapeutic implications according to distinct GIST molecular subtypes.

**Abstract:**

Gastrointestinal stromal tumors (GISTs) are malignant mesenchymal tumors arising from the intestinal pacemaker cells of Cajal. They compose a heterogenous group of tumors due to a variety of molecular alterations. The most common gain-of-function mutations in GISTs are either in the *KIT* (60–70%) or platelet-derived growth factor receptor alpha (*PDGFRA*) genes (10–15%), which are mutually exclusive. However, a smaller subset, lacking *KIT* and *PDGFRA* mutations, is considered wild-type GISTs and presents distinct molecular findings with the activation of different proliferative pathways, structural chromosomal and epigenetic changes, such as inactivation of the *NF1* gene, mutations in the succinate dehydrogenase (*SDH*), *BRAF*, and *RAS* genes, and also *NTRK* fusions. Currently, a molecular evaluation of GISTs is imperative in many scenarios, aiding in treatment decisions from the (neo)adjuvant to the metastatic setting. Here, we review the most recent data on the molecular profile of GISTs and highlight therapeutic implications according to distinct GIST molecular subtypes.

## 1. Introduction

Gastrointestinal stromal tumors (GISTs) are malignant mesenchymal tumors classified primarily as soft tissue sarcomas [1] and characteristically arise from the intestinal pacemaker cells of Cajal responsible for the gastrointestinal tract motility [2]. GISTs are the most frequent type of sarcomas and also the most frequent sarcoma of the gastrointestinal tract [3]. Most GISTs are sporadic, solitary tumors, and more are routinely found in the stomach or small intestines (60–65% and 20–35% of cases, respectively) [2]. Primary GISTs of the rectum can be found in 3–5% of cases, and an even smaller portion of cases can be found in the colon, esophagus, and peritoneal cavity. The most common metastatic sites are the liver and peritoneal cavity, mostly due to anatomic contiguity [4].

GISTs are more commonly diagnosed in an older subset of patients, and there is also a slight male predominance. A smaller portion of GISTs (approximately 2%) occur in pediatric patients [5] and can be associated with genetic syndromes such as neurofibromatiosis-1 (NF1), familial GISTs, the Carney Triad (CT) and Carney–Stratakis Syndrome (CSS) [4,6,7,8,9]. Pathological features of GISTs have already been demonstrated, and 95% of them express KIT (CD117) by immunohistochemistry [10] in addition to other markers such as DOG1 (95%), CD34 (60–70%), smooth muscle actin (SMA) (30–40%), and S100 protein (5%), which are more useful in diagnosing *KIT*-negative GISTs [11,12,13]. 

GISTs comprise a heterogenous group of tumors mostly due to a variety of molecular alterations. The most common gain-of-function mutations in GISTs are either in the *KIT* (60–70%) or platelet-derived growth factor receptor alpha (*PDGFRA*) genes (10–15%) [2], which are mutually exclusive [14]. However, a smaller subset, which lacks *KIT* and *PDGFRA* mutations, is considered wild-type GISTs and presents distinct molecular findings with activation of different proliferative pathways, structural chromosomal and epigenetic changes, such as inactivation of the *NF1* gene, mutations in the succinate dehydrogenase (*SDH*), *BRAF*, and *RAS* genes, and also NTRK fusions [15]. Currently, molecular evaluation of GISTs is imperative in many scenarios.

Classically known to harbor resistance to cytotoxic chemotherapies, GISTs are considered a mark of the advances in precision medicine after the success of tyrosine kinase inhibitors (TKIs), such as imatinib, in the treatment of *KIT*-positive advanced GISTs in 2002 [16]. However, treatment resistance is still a pertinent issue. As novel mutations and further insights into the underlying molecular signature of GISTs were uncovered, new lines of therapy presented optimistic results with targeted therapy [17]. Nonetheless, there is still a portion of GISTs that present treatment challenges. Therefore, it is important to accurately evaluate tumor pathology, including cell morphology and immunohistochemistry (accessing positiveness for KIT and DOG1), and the molecular profile [3] in order to improve the prediction of the biological behavior of the tumor and define the prognosis, as well as treatment response. Here, we review the most recent data on the molecular profile of GISTs and the systemic therapeutic implications. 

## 2. Molecular Classification of GISTs

Until recently, GISTs were classified into two main groups: *KIT/PDGFRA* mutated GISTs (85% of cases) and wild-type GISTs. However, rare mutations or fusions have been described among the latter, which made the “wild-type” description less precise. Amongst the wild-type *KIT/PDGFRA* GISTs, subgroups with well-defined molecular features have been described, including those harboring mutations in the *BRAF*, *RAS*, or *NF1* genes [18,19]. GISTs with mutations in *BRAF/RAS* or *NF1* were once referred to as the *RAS*-pathway (*RAS*-P) mutant GISTs. In addition, around a third of wild-type *KIT/PDGFRA* GISTs demonstrate the loss-of-function of the succinate dehydrogenase complex (*SDH*), manifested by the loss of subunit B (SDHB) protein expression [19]. In 2015, an Italian group proposed that *KIT/PDGFRA/SDH/RAS-P* wild-type GISTs should be designated as quadruple wild-type GISTs [19].

Currently, there has been an effort to subdivide GISTs into two major groups according to SDH competency by SDHB immunohistochemistry [7,20]. SDH is a heterotetrameric enzyme complex located in the inner mitochondrial membrane and participates in the Krebs cycle. Subunit A (SDHA) is responsible for converting succinate into fumarate, while subunit B (SDHB) participates in the electron transport chain for the oxidation of ubiquinone to ubiquinol. Subunits C and D (SDHC and SDHD) are membrane-anchoring subunits [7]. Loss of any SDH subunit renders the complex inactive and leads to loss of detectable SDHB by immunohistochemistry.

The SDH-competent GIST group is by far the most common one. It includes the large proportion of patients who harbor either KIT or PDGFRA mutations, but also GISTs with less frequent mutations (i.e., BRAF, NF1, HRAS, NRAS), with even more rare mutations (i.e., ARID1A, ARID1B, ATR CBL, FGFR1, KRAS, LTK, MEN1, PARK2, PIK3CA, SUFU, and ZNF217) and those with structural chromosomal changes (i.e., FGFR1-HOOK3, FGFR1-TACC1, ETV6–NTRK3, KIT-PDGFRA, and PRKAR1B-BRAF) [21].

Conversely, the group of *SDH*-deficient GISTs includes wild-type GISTs in association with CT, CSS, and sporadic pediatric and young adult GISTs.

Given the different subtypes of GISTs, we recommend the following stepwise fashion of testing (Figure 1) whenever systemic treatment is indicated and next-generation sequencing (NGS) is not promptly available.

### 2.1. SDH-Competent GIST with either KIT or PDGFRA Mutation

*KIT* codifies the c-Kit protein (also known as stem cell factor (SCF) receptor or CD117), a receptor tyrosine kinase (RTK) with five Ig-like extracellular (EC) domains (therefore, class III RTK) with 145 kDa and 976 amino acid residues in normal splicing [22]. In humans, four isoforms of c-Kit have been described so far, with the GNNK-negative isoform showing greater kinase activity and downstream signaling. The first three EC domains are essential for SCF binding and the other two for receptor homodimerization. The EC domains are connected by a single spanning transmembrane helix to the intra-cellular juxtamembrane (JM) domain, which works as a modulator of the signal transduction. The tyrosine kinase domain is split in two functional lobes (N-terminal and C-terminal), and in the enzyme inactive state, the JM domain inserts between the two lobes, sterically blocking the conformational changes needed for the activation loop of the catalytic unit to assume its active form. Finally, following the C-terminal lobe of the tyrosine kinase domain is the C-terminal intra-cellular tail. Upon dimeric SCF binding to adjacent c-Kit molecules, reorientation of D4-5 EC domains leads to receptor homodimerization. The following conformational changes lead to sequential trans-phosphorylations in the JM domain, activation loop, kinase insert region, and C-terminal tail [22]. These phosphorylated tyrosine residues serve as substrate docking sites for signal transduction. The following signaling pathways are activated by c-Kit: mitogen-activated protein kinase (MAP kinase), phosphatidylinositol 3-kinase/protein kinase B (PI3K/AKT), phospholipase C gamma (PLC-gamma), Janus kinase/signal transducer and activator of transcription (JAK/STAT), and Scr kinase pathways [22,23]. The specific phosphorylated tyrosine residues determine which signaling pathways are activated. 

Primary *KIT* mutations occur in roughly 70% of GISTs [24]. These gain-of-function mutations lead to constitutive activation of the receptor and are commonly found in two hot spots (Figure 2): membrane-proximal EC domain (exon 9; 9–10% of all GISTs) and intra-cellular JM domain (exon 11; 60% of all GISTS). GISTs with exon 11 mutations can occur throughout the gastrointestinal tract, and those with exon 9 mutations arise primarily in the small or large bowel [2]. Importantly, the prognosis is dependent on the specific type of mutation within exon 11, with tumors displaying deletions in codons 557 and 558 presenting more aggressive clinical behavior. Primary mutations in exons 8 (membrane-proximal EC domain), 13 (ATP-binding site), and 17 (activation loop) are rare (<1% of all GISTs) [24].

Another important and common molecular alteration and drive mutation in GISTs is *PDFGA* mutation. As the *KIT* gene, *PDGFRA* is located on chromosome 4, encodes a type III RTK, and is homologous to KIT, presenting a similar structure and downstream pathways, such as *RAS/RAF/MAPK* and *PI3K* [14,25,26]. Importantly, *KIT* and *PDGFRA* mutations are considered to be mutually exclusive. *PDGFRA* mutations, once present, trigger a ligand-independent phosphorylation and uninhibited cellular proliferation [15]. 

*PDGFRA* mutations are mainly encountered in exon 18 (activation loop of the receptor; 13–14%) and less frequently in exon 12 (JM domain; 0.6–2%) or exon 14 (ATP-binding site; 1.4%) [2] (Figure 2). The *PDGFRA* exon 18 D842V missense mutation can be found in around 50–70% of *PDFRA*-mutated GISTs and 8% of all GISTs [27] and promotes the stabilization of the kinase conformation in its active form, therefore conferring resistance of this molecular subtype to the most commonly used treatments [28,29]. The majority of *PDFGRA*-mutated GISTs arise from the stomach (15–18%) and the small intestine (5–7%). They classically have a more indolent behavior [2] and favorable prognosis compared to the more commonly encountered *KIT* mutations. 

### 2.2. Therapeutic Implications of GIST with KIT or PDGFRA Mutations

The main therapeutic strategy to treat GISTs with either *KIT* or *PDGFRA* mutations has been the use of TKIs, especially imatinib. Initially developed to target the *BCR-ABL* translocation of chronic myelogenous leukemia [30], it was shown to be active against most GISTs [31]. Thus, imatinib is considered one of the first targeted therapies for the treatment of solid tumors. 

The initial treatment of high-risk (>50% recurrence risk) GISTs in the adjuvant setting is considered to be a daily 400 mg dose of imatinib for a period of ≥3 years [32,33,34]. The recommended duration of treatment was defined based on evidence that, with 3 years compared with 1 year of therapy, recurrence-free survival (RFS) at 5 years was 71.1% versus 52.3% and overall survival (OS) was 91.9% versus 82.3%, respectively [32]. However, studies have been conducted aiming to define the appropriate duration of adjuvant treatment with imatinib and possibly expanding the duration for a period of 5 years [35], and yet, other studies are underway in order to determine the efficacy of extending the duration of adjuvant treatment (NCT02413736, NCT02260505). 

In the advanced/metastatic disease setting, imatinib is started at a 400 mg daily dose, thus achieving response rates and progression-free survival (PFS) rates of up to 40 months and 70%, respectively, in patients with the most common *KIT* and *PDGFRA* mutations [36,37,38,39,40]. However, regardless of such outcomes with first-line imatinib, subsequent lines of therapy present only modest benefits, and wild-type *KIT/PDGFRA* GISTs typically present with early progression with imatinib [36,41]. 

Tumors with deletions of codons 557 and 558 present more aggressive clinical behavior than those with exon 11 missense mutations. However, such mutations display significant responses to imatinib. Furthermore, patients’ genetics seems to modify the chance of tumor response to imatinib. Among patients with exon 11 mutant GISTs, differences in clinical response to imatinib have been shown to be associated with single-nucleotide polymorphisms in genes related to the absorption, distribution, metabolism, and excretion (ADME) process [42]. GISTs with exon 9 typically present lower response to imatinib when compared to those with exon 11 mutations. Nonetheless, in the metastatic setting, this can be partially circumvented by doubling the dose of imatinib (800 mg/day). Data from randomized trials suggest that the higher dose of imatinib is associated with an improved objective response rate (47 vs. 21%) and PFS (HR = 0.57; *p* = 0.017) [39]. While some GISTs with primary exon 13 mutations are sensitive to imatinib, most primary exon 17 mutations confer primary resistance do imatinib [2]. 

Most *PDGFRA* mutations are sensitive to imatinib therapy (exons 12, 14, and 18). However, *PDGFRA* exon 18 D842V mutation can also present with primary resistance to imatinib [41], as well as other TKIs [14,24,43]. Yet, due to the more indolent nature of this GIST subtype, patients may present with stable disease while on therapy [40].

Of note, around 10% of patients diagnosed with advanced GIST can present with primary resistance to imatinib with disease progression within the first 6 months of treatment. This primary resistance to treatment is linked to the GIST genotype [36,38,39,43] and is found to be more frequent in *KIT* exon 9 mutations and in wild-type *KIT/PDGFRA* GISTs. 

Acquired *KIT* and *PDGFRA* mutations occur in up to 90% of patients with metastatic GIST treated with imatinib and are the main cause of secondary resistance [44]. In most situations, imatinib therapy induces tumor volume reduction upon inducing cellular apoptosis. However, a fraction of tumor cells enters a quiescent, non-proliferative state and may acquire secondary genetic mutations, rendering such cells imatinib-resistant, leading to disease progression. Such mutations confer an evolutionary advantage under the selective pressure of imatinib exposure and cluster into two hot regions of the *KIT* gene: the ATP-binding pocket (exons 13 and 14 of *KIT* and exon 14 of *PDGFRA*) and the activation loop (exons 17 and 18 of *KIT* and exon 18 of *PDGFRA*) [2,45,46,47]. Therefore, treatment sequencing with other TKIs becomes necessary to overcome disease progression due to such mutations. c-Kit molecules with secondary mutations in the ATP-binding pocket are relatively sensitive to sunitinib [48], ponatinib [49], ripretinib [17,50], and avapritinib [51,52], but not to regorafenib or sorafenib [2], as shown in Table 1. Conversely, receptors carrying acquired mutations in the activation loop can be targeted with regorafenib [53], sorafenib [54], nilotinib [55], ponatinib [49], dasatinib (NCT01643278), ripretinib [50], or avapritinib [51]. The one exception is the D816V mutation in exon 17, which is resistant to all TKIs with the exception of ponatinib, ripretinib, and avapritinib [44,56,57]. 

The *PDGFRA*-mutant GISTs’ resistance mechanism profile is less detailed compared to *KIT*, and little is known beyond primary or secondary *PDGFRA* D482V mutations [58]. However, D842V has been shown to be homologues to the *KIT* exon 17 D816V mutation [24] and, as such, demonstrates a similar treatment sensitivity profile with resistance to the majority of the approved TKIs for the treatment of advanced GISTs, harboring a very poor prognosis. Yet, in 2020, the NAVIGATOR trial demonstrated the activity of avapritinib, a fourth-generation TKI designed specifically to inhibit *PDGFRA* D842V in advanced GISTs with such mutation with an objective response rate of 84% (7% complete responses) [51]. However, resistance mechanisms to avapritinib in *PDGFRA* D842V mutated GISTs have been demonstrated and cause avapritinib binding blockage due to substitutions in exons 13, 14, and 15 (V658A, N659K, Y676C, and G680R) [58].

### 2.3. SDH-Competent GIST without KIT and PDGFRA Mutation

#### *NF1*-Mutant GIST

Neurofibromatosis 1 (NF1) is an autosomal dominant disorder caused by inherited or de novo germline mutations within the *NF1* tumor suppressor gene. It is clinically characterized by multiple café-au-lait macules, intertriginous freckling, multiple cutaneous neurofibromas, plexiform neurofibromas, malignant peripheral nerve sheath tumors and, sometimes, learning disability or behavioral problems [59,60]. The product of the *NF1* gene is the large and multifunctional cytoplasmic protein neurofibromin, which belongs to a family of guanosine-triphosphate-hydrolase (GTPase)-activating proteins that stimulate intrinsic GTPase activity in the Ras family [59]. 

Patients with germline inactivation of the *NF1* gene have a propensity to develop several tumors through acquired inactivation of the normal *NF1* allele, and it has been estimated that 7% of individuals with NF1 develop GIST during their lifespan, which corresponds to a 34-fold higher risk than the average population [59,61]. NF1-associated GISTs, which correspond to 1% of all GISTs, are commonly multicentric and predominantly located in the small intestine [62,63] as shown in Table 2. Although most cases of NF1-related GISTs are germinative, somatic inactivating *NF1* mutations may also occur as an oncogenic mechanism for a small subset of wild-type GISTs [62].

NF1-related GISTs typically express KIT, DOG1, and SDHB by IHC [64]. They also frequently demonstrate loss of heterozygosity (LOH) at 14q and 22q, which may contribute to the early phase of tumor development [65]. In addition, genomic profiling studies of adult GISTs have revealed that *NF1* alterations are enriched in GISTs located at the duodenal–jejunal flexure. These tumors may harbor concurrent activating *KIT* and/or inactivating Notch pathway mutations [63].

### 2.4. Therapeutic Implications of NF1-Mutant GISTs

Since most *NF1*-mutant GISTs do not harbor KIT mutations, they are typically insensitive to imatinib and surgery remains the mainstay therapy for these patients. Although, so far, there are no data on target-specific drugs for NF1-related GISTs, MEK inhibitors have pointed towards clinical efficacy in other NF1-associated tumors. Given the key role of abnormal RAF/MEK/ERK pathway signaling in neurofibromas, a phase I study of selumetinib (a MEK inhibitor) in children/young adults with NF1-related inoperable plexiform neurofibromas has demonstrated a median maximal decrease of 24% in the volume of the tumors [66]. Currently, a phase II trial with a different MEK inhibitor is ongoing for adolescents and young adults with NF1-associated plexiform neurofibromas (NCT02096471).

### 2.5. BRAF Mutated GIST

*BRAF* mutations at the exon 15 *V600E* hotspot are encountered in approximately 4–13% of adult wild-type GISTs and are almost mutually exclusive of *KIT/PDGFRA* mutations [67,68,69]. They are more frequently found in the small bowel and have variable clinical behavior [67]. Phenotypically and morphologically, *BRAF*-mutant GISTs are similar to *KIT/PDGFRA* positive GISTs [67,68,69]. More recently, *BRAF* fusions have been described in GISTs [70].

### 2.6. Therapeutic Implications of BRAF Mutant GISTs

*BRAF V600E* mutation in GISTs has been shown to confer resistance to imatinib and sunitinib [71]. Since *BRAF V600E* mutations are currently considered an agnostic genetic alteration and seem to respond to BRAF inhibitors regardless of the primary tumor type, we recommend a BRAF inhibitor with or without a MEK inhibitor to patients with unresectable or metastatic *BRAF* mutant GISTs. The combination of dabrafenib (BRAF inhibitor) and trametinib (MEK inhibitor) has recently received FDA approval for any *BRAF* mutant tumor, regardless of the origin site. The first case of a patient with a *BRAF* mutant GIST who responded during 8 months to dabrafenib was reported in 2013 [72]. This patient had already failed to receive benefit from imatinib, sunitinib, and a MEK inhibitor.

Interestingly, a complete radiological response with first-line regorafenib was reported in a 51-year-old woman with a *BRAF*-mutated GIST [73]. Regorafenib has a wide spectrum of action, with *MAPK* signaling pathway blockade at different levels. There is an ongoing phase II, single-arm, trial evaluating regorafenib in the first-line setting for advanced *KIT/PDGFRA* wild-type GISTs (NCT02638766).

The role of kinase inhibitors targeting *BRAF* is yet unknown in the adjuvant setting for this subset of patients. Therefore, we recommend no systemic therapy following R0 resections, even for those with high-risk *BRAF* mutant disease.

### 2.7. GISTs with NTRK Fusion

TRK fusions arise from aberrant rearrangements of neurotrophic receptor kinase (NTRK) genes 1–3 with several gene partners. They are constitutively active ligand-independent oncogenic drivers in different solid tumors, including GISTs. Until recently, only ETV6-NTRK3 gene fusions were described in GISTs. However, NTRK 1 fusion has been recently reported [74]. 

### 2.8. Therapeutic Implications of GISTs Harboring NTRK Fusions

NTRK fusions are considered an agnostic feature that renders tumors sensitive to TRK inhibitors independently of the primary cancer. In the metastatic setting, larotrectinib, a TRK inhibitor, demonstrated great activity against solid tumors harboring NTRK fusions [75]. In a pooled analysis from three phase I/II clinical trials, 159 patients with different tumor types harboring NTRK fusions were treated with larotrectinib. The objective response rate was 79%. Indeed, four patients with GISTs were included, and all of them achieved an objective response [75]. In another pooled analysis from three different studies of individuals with NTRK fusions, three patients with GIST were treated with larotrectinib and all responded, including one with complete response [76].

### 2.9. GISTs with FGFR Pathway Alterations

Fibroblast growth factor receptor (FGFR) fusions, mutations, and ligand overexpression represent the most common molecular alterations in quadruple WT GISTs, suggesting a possible pathogenetic role [77]. 

### 2.10. Therapeutic Implications of FGFR-Altered GISTs

Many drugs targeting FGFR are already in use in clinical practice (such as regorafenib) or have been tested in clinical trials for GISTs (dovitinib, masitinib, ponatinib, and pazopanib), but none so far specifically for FGFR-altered GISTs [75,78,79,80,81,82,83,84,85,86]. 

Although very promising, FGF/FGFR pathway inhibition in GISTs is still far from clinical practice.

### 2.11. Other Rare Mutations

KRAS-mutated GISTs are extremely rare, and the clinicopathologic features still lack complete comprehension [87]. Yet another extremely rare GIST subtype are the *PIK3CA* mutant GISTs. Currently, the most robust study involving imatinib-naïve GIST patients, 10 (mostly localized tumors) out of 529 GISTs harbored *PIK3CA* mutations, which were associated with larger and aggressive tumors [88] ]. Given the limited number of patients, further studies with longer follow up are required. 

### 2.12. SDH-Deficient GIST

Wild-type GISTs are the primary form of GIST in children (85%) and only occasionally occur in adults (15%) [89]. They are often localized in the stomach, multifocal, with lymph node involvement, indolent, and primarily affect young females [90]. Approximately half of pediatric WT-GISTs are SDH-deficient, which have loss-of-function of the SDH complex subunit (SDHA, SDHB, SDHC, or SDHD). This loss of function occurs either due to a mutation in one of the SDHx genes or due to epigenetic silencing [91,92]. 

The SDH complex is a mitochondrial complex that participates in the Krebs cycle in the conversion of succinate to fumarate (succinate reductase). Loss of the mitochondrial SDH complex through mutations in SDHA, SDHB, SDHC, or SDHD causes the accumulation of succinate, resulting in the overexpression of hypoxia-inducible factor (HIF) proteins and increased transcription of HIF-1a-regulated genes. It also leads to decreased DNA demethylation. Hence, an increase in DNA methylation and activation of insulin-like growth factor 1 receptor (IGF1R) and vascular endothelial growth factor receptor (VEGFR) are the molecular characteristics of SDH-deficient GISTs [93]. 

SDH-deficient GISTs have different clinical courses, molecular characteristics, prognoses, and therapeutic responses when compared to other GISTs. SDHA is the most commonly mutated (30%) in all SDH-deficient GISTs [94]. Immunohistochemical analysis for SDHB is appropriate to diagnose SDH deficiency in *KIT/PDGFRA* wild-type GISTs, as the absence of SDHB can indicate the deficiency of any of the SDHx genes. However, not all SDHB-immunonegative GISTs harbor an SDH gene mutation; these tumors may have other epigenetic and genetic defects in the SDH pathway [8]. 

Approximately 5% of patients with GISTs have genetic syndromes associated with the development of these tumors, which include primary familial GIST syndrome (heritable mutations in either the *KIT* or *PDGFRA* genes), NF1, CSS (SDHx genes), and CT (hypermethylation of the SDHC promoter). CSS, also referred to as the Carney–Stratakis dyad, is a rare inherited predisposition syndrome caused by germline mutations in the SDHB/C or D subunits, which presents with the dyad of GISTs and paragangliomas [95,96]. The Carney triad is an extremely rare nonhereditary syndrome consisting of GISTs, paragangliomas, and pulmonary chondromas caused by epigenetic inactivation (hypermethylation) of the *SDHC* gene with functional impairment of the SDH complex [95,97,98].

Patients diagnosed with SDH-deficient GISTs and those who have NF1 mutations should be referred for genetic counseling and appropriate germline testing. Furthermore, patients with SDH-deficient GISTs, or known *SDH* mutations, could have paraganglioma associated (Carney–Stratakis dyad or CT), and therefore, serum/urine catecholamine/metanephrine testing should be considered prior to surgery [99].

### 2.13. Therapeutic Implications of SDH-Deficient GISTs

An observational study evaluated the clinical and molecular features of *KIT/PDGFRA* WT-GISTs in a cohort (patients < 19 y with GIST or > 19 y with WT-GIST), which was characterized by IHC for SDHB, sequencing of *SDH* genes, determination of *SDHC* promoter methylation, as well as germline testing of *SDH* genes for some of them (consented). Among the 95 patients analyzed (70% female; median age: 23 years; 19% with syndromic GIST), three molecular subtypes were defined: 11 had SDH competency (which included mutations in *NF1*; *BRAF*; other rare mutations/fusions; or no identified), and 84 had SDH deficiency (75% due to *SDH* mutations and 25% due to *SDHC* promoter hypermethylation). *SDH* mutations were often germline (82%) among 38 tested patients. In this cohort, despite the non-standardized retrospective assessment of treatment, only 1 of 49 patients treated with imatinib and 4 of 38 patients treated with sunitinib had an objective response [7].

SDH-deficient GIST is an indolent disease, and most patients have a good median survival of approximately 10 years even though with disease progression [100]. This subtype of GISTs is known to be generally unresponsive to TKIs. In the analysis of molecular alterations associated with the treatment benefit of adjuvant imatinib in the ACOSOG Z9001 study, *KIT/PDGFRA* WT tumors were uncommon (9 in the placebo and 6 in the imatinib group). As a result, the therapeutic impact of adjuvant imatinib could not be considered definitive in this subset and remains controversial [101]. The National Comprehensive Cancer Network (NCCN) guidelines version 2.2022 [99] and the European Society for Medical Oncology (ESMO) guidelines 2021 [3] do not support routine clinical use of imatinib adjuvant for WT-GISTs. In the advanced scenario, the best way to treat these patients is not established, and personalized treatment is recommended [3]. Although refractory to imatinib, they may have some responsiveness to sunitinib [102], regorafenib [103], Pazopanib [85], Nilotinib, and Linsitinib [104]. However, the data on TKIs in patients with SDH-deficient GISTs is extremely limited. Referral to a clinical trial is a reasonable consideration for them. 

Molecular characterization demonstrated that temozolomide (TMZ) elicits DNA damage and apoptosis in SDH-deficient GISTs with a 40% rate of objective responses among five patients treated with this alkylating agent [105]. Based on this early efficacy data, a phase II study of single-agent TMZ (NCT03556384) in patients with SDH-mutant GISTs is ongoing. 

Other agents are under study in clinical trials, including rogaratinib (potent and selective pan-FGFR inhibitor/NCT04595747), vandetanib (potent VEGFR inhibitor/NCT02015065), and guadecitabine (DNA methyl transferase inhibitor/NCT03165721). In addition, DNA hypomethylating agents are being evaluated in SDH-deficient GISTs, which may be associated with DNA hypermethylation [106]. 

## 3. Conclusions

Recent advances in molecular profiling of GISTs brought significant value to prognostication and therapeutic decisions in clinical practice. New drugs have been approved for advanced GISTs with either the *KIT* or *PDGFRA* mutation. Nonetheless, there are significant unmet needs for the subset of patients with *KIT/PDGFRA* wild-type GISTs.

## Figures and Tables

**Figure 1 cancers-14-05330-f001:**
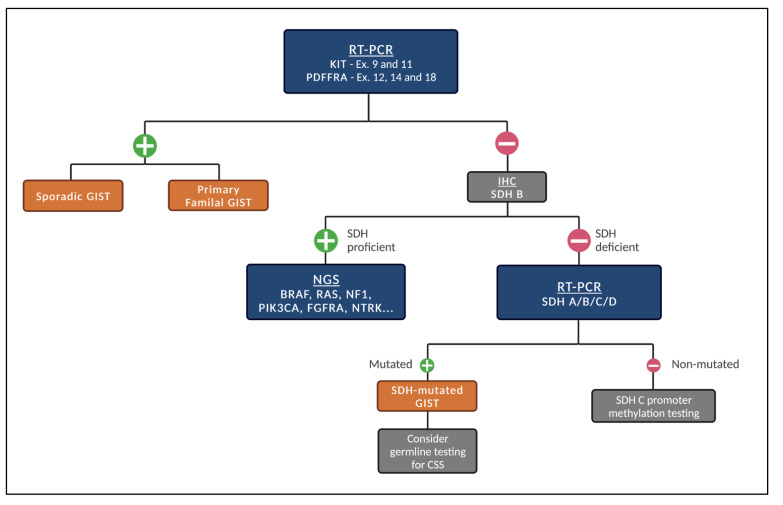
Proposed stepwise molecular testing for GISTs (when NGS is not promptly available). Legend: Initial test for the presence of *KIT* and *PDGFRA* mutations. If negative, test for SDH status by IHC. If SDH-proficient, patients should be tested for other genetic alterations. If SDH-deficient, test for mutations in *SDH A/B/C/D*. When mutated, consider germline testing for CSS. If not mutated, consider SDH C promoter methylation testing.

**Figure 2 cancers-14-05330-f002:**
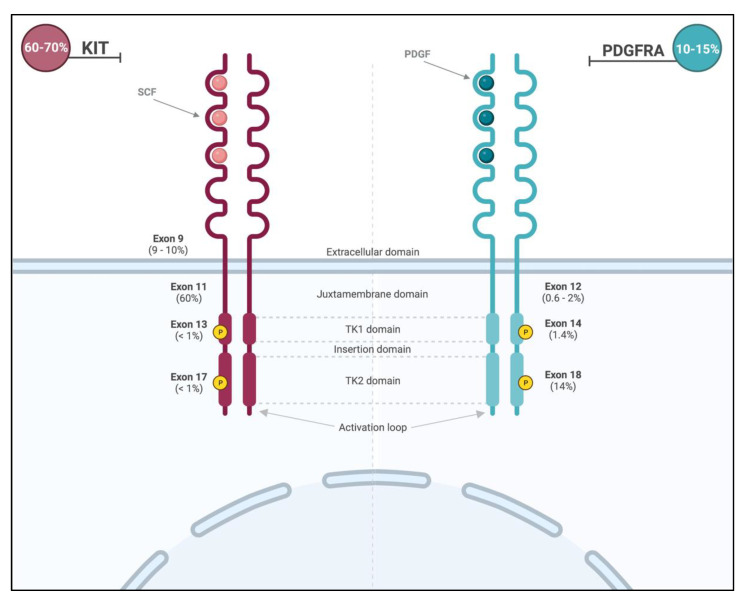
Location and frequencies of most common KIT and PDGFRA mutations.

**Table 1 cancers-14-05330-t001:** Sensitivity profile of different molecular subtypes of *KIT* or *PDGFRA* GISTs.

Molecular GIST Sub-Type	Treatment				
	Imatinib [14,31,43]	Sunitinib [45]	Regorafenib [53]	Ripretinib [50]	Avapritinib [51]
*KIT* Mutations					
Exon 9	✓	-	-	-	-
Exon 11	✓	-	-	-	-
Exon 13 (V654)	✕	✓	✕	✓	-
Exon 14 (T670)	✕	✓	✓	✓	-
Exon 17 (D816)	✕	✕	✓/✕	✓/✕	-
Exon 17 (D820)	✕	✕	✓	✓	-
Exon 17 (N822)	✕	✕	✓	✓	-
Exon 17 (Y823)	✕	✕	✓	✓	-
Exon 18 (A829)	✕	✕	✓	✓	-
*PDGFRA* Mutations					
Exon 12	✓	✓	✓	✓	✓
Exon 13	✕	✕	✕	✕	✕
Exon 14	✕	✕	✕	✕	✕
Exon 15	✕	✕	✕	✕	✕
Exon 18 (D842V)	✕	✕	✕	✕	✓
Exon 18 (Non-D842V)	✓	✓	✓	✓	✓

The checkmarks indicate drug sensitivity, and the crosses indicate drug resistance.

**Table 2 cancers-14-05330-t002:** GIST molecular subtypes and characteristics.

Alterations	Characteristics	Systemic Therapeutic Options
**KIT Mutations (60–70%)**		
**Exon 9** (9–10%)	Small or large intestine	First-line recommended treatment: Less sensitive to imatinib 400 mg/daily dose, with higher responses to 800 mg/daily dose.
**Exon 11** (60%)	Gastrointestinal tract; Del 557 and 558—more aggressive	First-line recommended treatment: Imatinib 400 mg/daily dose.
**Exon 13** (less than 1%)	All sites	First-line recommended treatment: Imatinib 400 mg/daily dose—sensitivity is low; sensitivity improves with other TKIs such as regorafenib and sunitinib.
**Exon 17** (less than 1%)	All sites	First-line recommended treatment: Imatinib 400 mg/daily dose—usually presents primary resistance. Sensitivity improves with other TKIs such as regorafenib and sunitinib. *D816V mutation*-resistant to all TKIs with the exception of ponatinib, ripretinib, and avapritinib.
**PDGFRA Mutations (10–15%)**		
**Exon 12** (up to 2%)	Gastric (15–18%) and small intestine (5–7%). More indolent behavior and favorable prognosis	First-line recommended treatment: Imatinib 400 mg/daily dose
**Exon 14** (less than 2%)		
**Exon 18 (non-D842V)** (1–2%)		
**Exon 18 (D842V)** (9–10%)		First-line recommended treatment: Primary resistance to imatinib therapy. Avapritinib is the preferred regimen. *Homologous to**D816V mutation*—resistant to all TKIs with the exception of ponatinib, ripretinib, and avapritinib.
**KIT and PDGFRA wild-type–SDH-competent**		
***NF1* mutation** (1%)	Small intestines and multicentric	First-line recommended treatment: Typically, insensitive to imatinib; surgery is the primary treatment. Possible clinical efficacy of MEK inhibitors.
***RAS* mutation** (rare)	Unknown	Not sensitive to usual TKIs
***BRAF* mutations** (4–13%)	Small intestines and variable clinical behavior Phenotypically and morphologically, similar to *KIT/PDGFRA*-positive GISTs	First-line recommended treatment: iBRAF ± iMEK.
**Other mutations** (rare)	NTRK translocations—unknown	First-line recommended treatment: Specific inhibitors.
**KIT and PDGFRA wild-type–SDH-deficient**		
***SDHA, SDHB, SDHC,* or *SDHD* mutations** (<3%) *Carney–Stratakis syndrome*	Gastric and small intestine Children, adolescents, and young adults; lymph node involvement, indolent disease	Generally resistant to imatinib; can present sensitivity to anti-angiogenic TKIs
**Loss of SDHB expression** (<1%) *Carney triad*		

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
