# Peer review of "Current Molecular Profile of Gastrointestinal Stromal Tumors and Systemic Therapeutic Implications"

_cancers, 2022, doi:10.3390/cancers14215330_

Round 1

Reviewer 1 Report

Good review on molecular profile of GIST, however I have major remarks which should be included before possible publication:

1.       Fig 1 in the title comprise the scenario when NGS is unavailable but it starts from NGS technique as the first step, I think that this Figure should be corrected as still in many centers the molecular profiling of GIST starts from step by step PCR with KIT evaluation followed by PDGFRA.

2.       In SDH-defficient GIST succinate dehydrogenase deficiency is related to hypermethylation of the genes involved in chromatin cell differentiation, thus the use of DNA hypomethylating agents is under investigation for these tumours, it should be better described in the text.

3.       I would suggest to add the table summarizing the molecular subtypes of GIST with short characteristics and systemic therapies.

4.       I suggest to add in the title “… therapeutic implications for systemic therapies” as authors did not include the locoregional treatment in their considerations.

5.       Gene names should be written in Italics

6.       The text requires linguistic corrections.

Author Response

Dear Reviewer 1,

Thank you so much for your constructive advices. Here are our responses:

  1. Fig 1 in the title comprise the scenario when NGS is unavailable but it starts from NGS technique as the first step, I think that this Figure should be corrected as still in many centers the molecular profiling of GIST starts from step by step PCR with KIT evaluation followed by PDGFRA.

We corrected fig 1. Instead of calling it narrow NGS, we changed it to narrowed gene panel (KIT and PDGFR).

  1. In SDH-defficient GIST succinate dehydrogenase deficiency is related to hypermethylation of the genes involved in chromatin cell differentiation, thus the use of DNA hypomethylating agents is under investigation for these tumours, it should be better described in the text.

We added it.

  1. I would suggest to add the table summarizing the molecular subtypes of GIST with short characteristics and systemic therapies.

Alterations

Characteristics

Systemic Therapeutic Options

KIT Mutations (60-70%)

Exon 9 (9-10%)

Small or large intestine

First line recommended treatment: Less sensitive to imatinib 400mg/daily dose, with higher responses to 800mg/daily dose.

Exon 11 (60%)

Gastrointestinal tract;
Del 557 and 558 - more aggressive

First line recommended treatment: Imatinib 400mg/daily dose.  

Exon 13 (less then 1%)

All sites

First line recommended treatment: Imatinib 400mg/daily dose – sensitivity is low, sensitivity improves with other TKIs such as regorafenib and sunitinib.

Exon 17 (less then 1%)

All sites

First line recommended treatment: Imatinib 400mg/daily dose – usually present primary resistance.

Sensitivity improves with other TKIs such as regorafenib and sunitinib.
* D816V mutation - resistant to all TKIs with exception of ponatinib, ripretinib, and avapritinib

PDGFRA Mutations (10-15%)

Exon 12 (up to 2%)

Gastric (15-18%) and small intestine (5-7%).

More indolent behavior and favorable prognosis

First line recommended treatment: Imatinib 400mg/daily dose

Exon 14 (less then 2%)

Exon 18 (non-D842V) (1-2%)

Exon 18 (D842V) (9-10%)

First line recommended treatment: primary resistance to imatinib therapy

Avapritinib is the preferred regimen.

*Homologous to D816V mutation - resistant to all TKIs with exception of ponatinib, ripretinib, and avapritinib

KIT and PDGFRA wild-type – SDH-competent

NF1 mutation (1%)

Small intestines and multicentric

First line recommended treatment: typically insensitive to imatinib; surgery is the primary treatment

Possible clinical efficacy of MEK inhibitors.

RAS mutation (rare)

Unknown

Not sensitive do usual TKIs

BRAF mutations (4-13%)

Small intestines and variable clinical behavior.

Phenotypically and morphologically, similar to KIT/PDGFRA positive GISTs

First line recommended treatment: iBRAF ± iMEK;

Other mutations (rare)

NTRK translocations – unknown

First line recommended treatment: specific inhibitors

KIT and PDGFRA wild-type – SDH-deficient

SDHA, SDHB, SDHC or SDHD mutations (< 3%)

Carney Stratakis syndrome

Gastric and small intestine

Children, adolescents, and young adults; lymph node involvement, indolent disease

Generally resistant to imatinib; can present sensitivity to anti-angiogenic TKIs

Loss of SDHB expression (< 1%)

Carney triad

  1. I suggest to add in the title “… therapeutic implications for systemic therapies” as authors did not include the locoregional treatment in their considerations.

We have added it, accordingly.

  1. Gene names should be written in Italics

We have corrected them, accordingly.

  1. The text requires linguistic corrections.

       We have reviewed the entire manuscript and corrected some English mistakes.

Reviewer 2 Report

A clearly written comprehensive review of the current molecular classification and treatment options of GIST.

Reference # 32 should be replaced with the original report on the effect of Imatinib on GIST (Joensuu et al. NEJM 2001, 344) 

Author Response

Dear Reviewer 2,

Thank you for your advice. We have corrected the reference accordingly.

Reviewer 3 Report

This manuscript is a useful review of the molecular profiling of GIST and the treatments based on these molecular alterations accordingly. However, several points need to be addressed before considering publication.

1. The term "narrow NGS" is confusing. Generally, for the single-gene exon testing, we would use RT-PCR. NGS is referred to as high-throughput technology. Even panel-targeted sequencing commonly includes tens or hundreds gene alterations. I do not think narrow NGS is suitable to apply in detecting specific hotspot mutations of only two genes, namely KIT and PDGFRa. 

2. Table 1 and the corresponding text, page 5, lines 197-203, need several references to being cited to enhance the evidence. 

3. "Therapeutic implications of BRAF-mutant GISTs" In this section, tissue-agnostic indication/data with dabrafenib plus trametinib should be added.

4.  In the SDH-deficient GIST section, there are other therapeutic applications such as chemotherapy (temozolomide) and other TKIs

5. In the SDH-proficient GIST section, some quadruple wild-type GISTs, although rare, harbored FGFR/FGF signaling, which also needs to be addressed as well as current development of FGFR inhibitor in this case . 

minor:

1. "Tumors with deletions of codons 557 and 558 present more aggressive clinical behavior than those with exon 11 missense mutations."  Please add a reference to this statement accordingly.

Author Response

  1. The term "narrow NGS" is confusing. Generally, for the single-gene exon testing, we would use RT-PCR. NGS is referred to as high-throughput technology. Even panel-targeted sequencing commonly includes tens or hundreds gene alterations. I do not think narrow NGS is suitable to apply in detecting specific hotspot mutations of only two genes, namely KIT and PDGFRa. Response: We changed it accordingly. Thank you.

2. Table 1 and the corresponding text, page 5, lines 197-203, need several references to being cited to enhance the evidence. Response: We added them accordingly. Thank you.

3. "Therapeutic implications of BRAF-mutant GISTs" In this section, tissue-agnostic indication/data with dabrafenib plus trametinib should be added. We added it accordingly. Thank you.

4.  In the SDH-deficient GIST section, there are other therapeutic applications such as chemotherapy (temozolomide) and other TKIs. We added them accordingly. Thank you.

5. In the SDH-proficient GIST section, some quadruple wild-type GISTs, although rare, harbored FGFR/FGF signaling, which also needs to be addressed as well as current development of FGFR inhibitor in this case . We added them accordingly. Thank you.

minor:

  1. "Tumors with deletions of codons 557 and 558 present more aggressive clinical behavior than those with exon 11 missense mutations."  Please add a reference to this statement accordingly. We added it accordingly. Thank you.

Reviewer 4 Report

The manuscript titled “Current Molecular Profile of Gastrointestinal Stromal Tumors and Therapeutic Implications” is very interesting and well written.

The topic is important.

The manuscript could be accepted after minor revisions

Please cite the following manuscript in the discussion: 

-       Ravegnini G et al. An exploratory study by DMET array identifies a germline signature associated with imatinib response in gastrointestinal stromal tumor. Pharmacogenomics J. 2019 Aug;19(4):390-400. doi: 10.1038/s41397-018-0050-4. Epub 2018 Sep 20. PMID: 30237583.

Author Response

Dear Reviewer,

Thank you for your inputs. We added this reference accordingly.

Round 2

Reviewer 1 Report

Thank you for corrected version, I am in favor to accept it as it is.